# Adversarial Vulnerabilities in Large Language Models for Time Series Forecasting

**Fuqiang Liu**[*]
McGill University
fuqiang.liu@mail.mcgill.ca

**Sicong Jiang**[*]
McGill University
sicong.jiang@mail.mcgill.ca

**Luis Miranda-Moreno**
McGill University
luis.miranda-moreno@mcgill.ca

**Seongjin Choi**[†]
University of Minnesota
chois@umn.edu

**Lijun Sun**[†]
McGill University
lijun.sun@mcgill.ca

## Abstract

Large Language Models (LLMs) have recently demonstrated significant potential in the field of time series forecasting, offering impressive capabilities in handling complex temporal data. However, their robustness and reliability in real-world applications remain under-explored, particularly concerning their susceptibility to adversarial attacks. In this paper, we introduce a targeted adversarial attack framework for LLM-based time series forecasting. By employing both gradient-free and black-box optimization methods, we generate minimal yet highly effective perturbations that significantly degrade the forecasting accuracy across multiple datasets and LLM architectures. Our experiments, which include models like TimeGPT and LLM-Time with GPT-3.5, GPT-4, LLaMa, and Mistral, show that adversarial attacks lead to much more severe performance degradation than random noise, and demonstrate the broad effectiveness of our attacks across different LLMs. The results underscore the critical vulnerabilities of LLMs in time series forecasting, highlighting the need for robust defense mechanisms to ensure their reliable deployment in practical applications.

## 1 Introduction

Time series forecasting plays a pivotal role in numerous real-world applications, ranging from finance and healthcare to energy management and climate modeling. Accurately predicting temporal patterns in the data is crucial for informed decision-making in these domains [1]. Recently, Large Language Models (LLMs), originally designed for Natural Language Processing (NLP) tasks, have demonstrated remarkable potential in handling time series forecasting challenges [2, 3, 4, 5, 6]. These models, including BERT [7], GPT [8, 9], LLaMa [10] and their successors, leverage their powerful attention mechanisms and vast pre-training on diverse datasets to capture intricate temporal dependencies, making them highly effective for complex forecasting tasks.

LLMs exhibit strong generalization capabilities across various types of time series data. Compared to traditional models like ARIMA[11] and Exponential Smoothing [12], as well as advanced deep learning models such as DNNs [13, 14, 15], and Transformer-based architectures[16, 17, 18, 19], LLMs excel in modeling long-term dependencies and capturing non-linear patterns within temporal sequences. This has resulted in impressive forecasting accuracy across applications ranging from energy consumption predictions to weather forecasting [20, 21].

---

[*]Co-first authors.

[†]Corresponding authors.

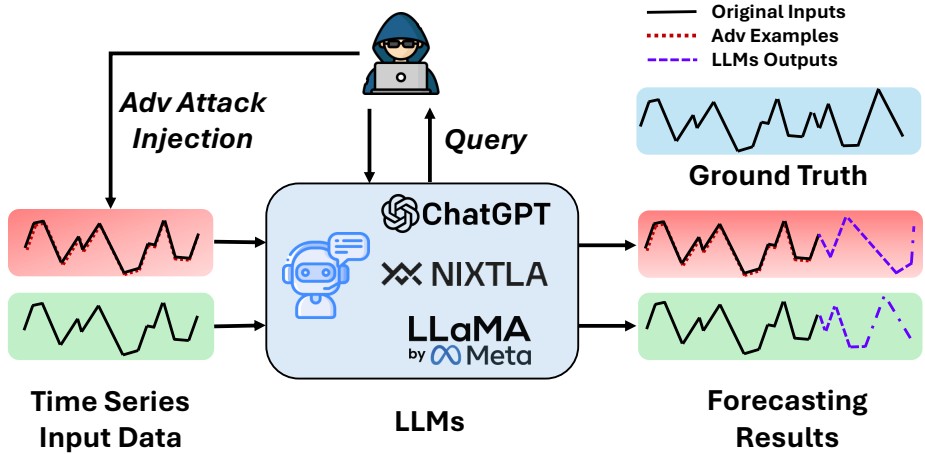

Figure 1: Adversarial Black-box Attack for LLMs in Time Series Forecasting

However, despite their success, the robustness and reliability of LLMs in real-world forecasting remain concerns, particularly their vulnerability to adversarial attacks is under-explored. Adversarial attacks introduce subtle, often imperceptible perturbations to input data, leading to significant and misleading changes in model predictions. While the susceptibility of machine learning models to such attacks has been well-explored in image processing and NLP domains [22, 23, 24], there is a noticeable gap in research on their impact on LLMs used for time series forecasting.

While adversarial attacks and defenses for deep neural networks have been extensively studied across various domains [25], executing adversarial attacks against LLMs in time series forecasting presents two significant challenges. First, to prevent information leakage, we cannot use ground truth values (i.e., future time steps) when attacking forecasting models. Second, LLMs must be treated as strict black-box systems due to the difficulty of accessing their internal workings and parameters.

In this paper, we address this gap by proposing a gradient-free black-box attack that transforms the output of LLM-based forecasting models into a random walk, while investigating the vulnerabilities of large language models in time series forecasting. As depicted in Figure 1, we demonstrate that even minimal attack perturbations can cause substantial deviations in LLMs' predictions. We evaluate two forms of LLM applications for time series forecasting, encompassing five sub-models, across five datasets from various real-world domains. Our findings reveal that LLMs, despite their advanced architectures, are indeed susceptible to adversarial manipulations in time series domain, leading to unstable and inaccurate forecasts. This underscores the urgent need to develop more robust LLMs that can withstand such attacks, ensuring their reliability in real-world applications.

In conclusion, this study contributes to the growing discourse on LLMs robustness by exposing their vulnerabilities to adversarial attacks in Time Series Forecasting. Our results highlight the necessity of addressing these vulnerabilities to advance the development of LLMs that are not only accurate but also resilient, thereby enhancing their practical utility in high-stakes environments.

## 2 Related Work

### 2.1 Adversarial Attacks in Time Series Forecasting

Adversarial attacks in time series forecasting have emerged as a crucial area of research, exposing vulnerabilities in forecasting models. Unlike adversarial studies in static domains, such as object recognition or time series classification, adversarial attacks on time series forecasting cannot leverage ground truth data for perturbation generation due to the risk of information leakage [26]. To address this challenge, surrogate techniques have been adopted [27], which bypass the need for labels, as is done in traditional adversarial attack methods like the Fast Gradient Sign Method [28]. Several studies have treated forecasting models as white-box systems to investigate the effects of adversarial attacks

on commonly used models in time series forecasting, such as ARIMA, LSTMs, and Transformer-based models [29, 30]. These studies demonstrate that even small perturbations can severely impact these models, resulting in inaccurate forecasts. However, evaluating the vulnerability of LLM-based forecasting presents a significant challenge, as internal access is typically restricted, requiring these models to be treated as black-box systems.

## 2.2 Adversarial Attacks on LLMs

Adversarial attacks on LLMs have gained increasing attention, focusing on how slight manipulations can significantly alter their outputs. These attacks are often classified into prompt-based attacks, token-level manipulations, gradient-based attacks, and embedding perturbations.

- **Jailbreak Prompting [31, 32]:** Crafted prompts that bypass LLM guardrails, inducing unintended or harmful outputs by exploiting unconventional phrasing.
- **Prompt Injection [33, 34, 35]:** Adversarial instructions embedded into benign prompts to manipulate LLM responses, highlighting their vulnerability to prompt manipulation.
- **White-box Gradient Attacks [25, 36]:** Using internal model parameters, attackers apply gradient-based methods to perturb inputs, significantly altering outputs with minimal changes.
- **Black-box Attacks [37]:** Query-based attacks without model access, using techniques like Zeroth-Order Optimization to craft adversarial examples by estimating gradients.
- **Embedding Perturbations [38, 39]:** Subtle changes to input embeddings disrupt LLM's internal representations, leading to erroneous outputs with minimal visible input alterations.

While extensive research has been conducted on attacks against LLMs at various levels, most of these focus on text-based manipulations. However, there's a significant gap in understanding how LLMs perform in non-textual tasks, particularly time series forecasting. In language tasks, attacks typically manipulate static text inputs, such as words or prompts, to exploit the LLM's understanding and induce specific outputs. However, time series forecasting involves dynamic, evolving data points, requiring attackers to introduce perturbations that maintain the sequence's natural flow and coherence.

## 3 Manipulating LLM-based Time Series Forecasting

### 3.1 Formulations of LLM-based Time Series Forecasting

LLMs have shown promising performance in time series forecasting by leveraging their ability to perform next-token prediction, a technique originally developed for text-based tasks [2, 6]. A typical LLM-based time series forecasting model, denoted as $f(\cdot)$, consists of two primary components: an embedding or tokenization module that encodes the time series data into a sequence of tokens, and a pre-trained LLM that autoregressively predicts the subsequent tokens. The embedding module translates the raw time series into a format suitable for the LLM, while the LLM captures the temporal dependencies and generates predictions based on its learned representations.

Let $\mathbf{X}_t \in \mathbb{R}^d$ denote $d$-dimensional time series at time $t$, where $x_{i,t} = [\mathbf{X}_t]_i$ represents the observation of the $i$-th component of the time series. Given a sequence of recent $T$ historical observations $\mathbf{X}_{t-T+1:t}$, a forecasting model, $f(\cdot)$, is employed to predict the future values for the subsequent $\tau$ time steps. The prediction is formulated as:

$$\hat{\mathbf{Y}}_{t+1:t+\tau} = f\left(\mathbf{X}_{t-T+1:t}\right), \tag{1}$$

where $\hat{\mathbf{Y}}_{t+1:t+\tau}$ denotes the predicted future values and $\mathbf{Y}_{t+1:t+\tau}$ represents the corresponding ground truth values. It is important to note that the prediction horizon is typically less than or equal to the historical horizon, i.e., $\tau \leq T$.

### 3.2 Threat model

Our objective is to deceive an LLM-based time series forecasting model into producing anomalous outputs that deviate significantly from both its normal predictions and the corresponding ground

truth, through the introduction of imperceptible perturbations. This adversarial attack problem can be framed as an optimization task as follows:

$$\max_{\rho_{t-T+1:t}} \mathcal{L}\left(f\left(\mathbf{X}_{t-T+1:t} + \boldsymbol{\rho}_{t-T+1:t}\right), \mathbf{Y}_{t+1:t+\tau}\right)$$
$$\text{s.t.} \quad \|\rho_i\|_p \leq \epsilon, i \in [t - T + 1, t], \tag{2}$$

where $\mathbf{X}_{t-T+1:t}$ denotes the clean input, $\mathbf{Y}_{t+1:t+\tau}$ denotes the true future values, and $\boldsymbol{\rho}_{t-T+1:t}$ denotes the adversarial perturbations. Our objective is to deceive an LLM-based time series forecasting model into producing anomalous outputs that deviate significantly from both its normal predictions and the corresponding ground truth, through the introduction of imperceptible perturbations. This adversarial attack problem can be framed as an optimization task as follows: The loss function $\mathcal{L}$ quantifies the discrepancy between the model's output and the ground truth, while $\epsilon$ constrains the magnitude of the perturbations under the $\ell_p$-norm, ensuring that the adversarial attack remains imperceptible.

Since the true future values $\mathbf{Y}_{t+1:t+\tau}$ are typically inaccessible in practical time series forecasting, they are replaced with the predicted values $\hat{\mathbf{Y}}_{t+1:t+\tau}$ generated by the forecasting model. Consequently, Eq. 2 is reformulated as

$$\max_{\boldsymbol{\rho}_{t-T+1:t}} \mathcal{L}\left(f\left(\mathbf{X}_{t-T+1:t} + \boldsymbol{\rho}_{t-T+1:t}\right), \hat{\mathbf{Y}}_{t+1:t+\tau}\right)$$
$$\text{s.t.} \quad \|\rho_i\|_p \leq \epsilon, i \in [t - T + 1, t]. \tag{3}$$

In practical applications, it is infeasible to access the full set of detailed parameters of an LLM, leading the attacker to treat the target model as a black box. Additionally, obtaining the entire training dataset is impractical, meaning the attacker does not have access to this data. The attacker's capabilities are summarized as follows: (i) **no access to the training data**, (ii) **no access to internal information of the LLM-based forecasting model**, and (iii) **be able to query the target model**.

## 4 Target Attack with Directional Gradient Approximation

Since the attacker has no access to the internal parameters of the LLM, it is not feasible to compute gradients and use them to solve the optimization problem presented in Eq. 3. This results in a gradient-free optimization problem. To address this, we propose a gradient-free optimization approach, referred to as targeted attack with **D**irectional **G**radient **A**pproximation (DGA), aimed at generating perturbations that can effectively deceive LLM-based time series forecasting models.

We first adjust our objective to focus on misleading the forecasting model into producing outputs that closely resemble an anomalous sequence, rather than simply deviating from its normal predictions. Accordingly, the optimization problem in Eq. 3 is reformulated as

$$\min_{\boldsymbol{\rho}_{t-T+1:t}} \mathcal{L}\left(f\left(\mathbf{X}_{t-T+1:t} + \boldsymbol{\rho}_{t-T+1:t}\right), \mathcal{Y}\right)$$
$$\text{s.t.} \quad \|\rho_i\|_p \leq \epsilon, i \in [t - T + 1, t], \tag{4}$$

where $\mathcal{Y}$ represents the targeted anomalous time series.

Supposing $\boldsymbol{\theta}_{t-T+1:t}$ denote a random small signal, the gradient, $\boldsymbol{g}_{t-T+1:t}$, which approximates the direction from the normal output to the targeted anomalous output, can be expressed as

$$\boldsymbol{g}_{t-T+1:t} = \frac{\mathcal{L}\left(\mathcal{Y} - f\left(\mathbf{X}_{t-T+1:t} + \boldsymbol{\theta}_{t-T+1:t}\right)\right) - \mathcal{L}\left(\mathcal{Y} - f\left(\mathbf{X}_{t-T+1:t}\right)\right)}{\boldsymbol{\theta}_{t-T+1:t}}. \tag{5}$$

Supposing $\ell_1$-norm is applied in Eq. 4, the magnitude of the perturbation is strictly constrained to be imperceptible. The perturbation, $\boldsymbol{\rho}_{t-T+1:t}$, can be computed from the approximated gradient, and the temporary adversarial example, $\mathbf{X}'_{t-T+1:t}$, is generated as

$$\mathbf{X}'_{t-T+1:t} = \mathbf{X}_{t-T+1:t} + \boldsymbol{\rho}_{t-T+1:t} = \mathbf{X}_{t-T+1:t} + \epsilon \cdot \text{sign}\left(\boldsymbol{g}_{t-T+1:t}\right), \tag{6}$$

where $\text{sign}\left(\cdot\right)$ denotes the signum function.

A time series forecasting model that produces Gaussian White Noise (GWN) as its output is considered to generate an anomalous prediction. Consequently, GWN can be utilized as the target sequence in

Eq. 6, formulated as $\mathcal{Y} \sim \mathcal{N}(\mu, \sigma)$, where $\mu$ and $\sigma$ represent the mean and the standard deviation, respectively. Empirically, the mean and standard deviation of the input data can be used to generate GWN. This results in a situation where a temporally correlated time series is misleadingly predicted as independent and identically distributed (i.i.d.) noise. This approach highlights the model's inability to preserve temporal correlations when subjected to adversarial perturbations, thereby reinforcing the effectiveness of the adversarial attack.

## 5    Experiments

### 5.1    Datasets

To evaluate the proposed DGA and gain a further understanding of the vulnerability of LLM-based forecasting, We conducted experiments using five widely recognized real-world datasets that cover a broad range of time series forecasting tasks:

- **ETTh1 and ETTh2 (Electricity Transformer Temperature Hourly) [16]**: These datasets consist of two years of hourly recorded data from electricity transformers, capturing temperature and power consumption variables.
- **IstanbulTraffic[2]**: This dataset contains hourly measurements of road traffic volumes across different sensors. It captures temporal dependencies related to traffic patterns, making it ideal for testing models on dynamic and fluctuating time series data.
- **Weather [16]**: This dataset comprises meteorological data, including variables such as temperature, humidity, and wind speed, recorded hourly. It provides a challenging forecasting task due to the inherent variability and complexity of weather patterns.
- **Exchange [40]**: This dataset consists of daily exchange rates from eight foreign countries—Australia, the United Kingdom, Canada, Switzerland, China, Japan, New Zealand, and Singapore—covering the period from 1990 to 2016.

These diverse datasets allow us to evaluate the robustness of LLMs across different types of temporal dynamics and forecasting challenges. In our experiments, 50% of the data is used for training, while the remaining data is split evenly: 25% for validation and 25% for testing. It should be noted that the attacker does not access either the training or validation part. We use a 96-step historical time window as input to the forecasting model, which predicts the subsequent 48-step future values.

### 5.2    Target Models

To assess the impact of adversarial attacks on LLMs for time series forecasting, we selected two state-of-the-art LLM-based forecasting models as baselines, which together represent two common forms of LLM application for time series tasks:

- **TimeGPT [41]**: A large model specifically pre-trained with a vast amount of time series data. TimeGPT uses advanced attention mechanisms and temporal encoding to capture complex patterns in sequential data, making it a leading LLM designed explicitly for time series forecasting. Its pre-training, which is conducted from scratch using vast amounts of time series data, allows it to serve as a robust and versatile tool for a wide range of time-dependent applications.
- **LLMTime [2]**: This model treats time series forecasting as a next-token prediction task, using LLM architectures like GPT and LLaMa. By converting time series data into numerical sequences, LLM-Time enables these models to apply their sequence prediction strengths to time series. To test the robustness of our adversarial attacks, we experimented with base models including GPT-3.5, GPT-4, LLaMa, and Mistral, assessing their resilience when adapted from natural language processing to time series forecasting.
- **TimeLLM [6]**: TimeLLM presents a novel approach for time series forecasting by adapting LLMs with reprogramming input time series data into textual representations that are more compatible with LLMs, allowing the models to perform time series forecasting tasks without altering their pre-trained structures. The key innovation is the Prompt-as-Prefix (PaP) technique, which augments input context to guide the LLM in transforming reprogrammed data into accurate forecasts.

Table 1: Results for univariate time series forecasting with a consistent input length of 96 and an output length of 48 across all models and datasets. A lower MSE or MAE indicates better prediction performance. The perturbation scale is set to 2% of the mean value of each dataset. Bold text highlights the worst performance for each dataset and model combination.

| Models | LLMTime w/ GPT-3.5 | | LLMTime w/ GPT-4 | | LLMTime w/ LLaMa 2 | | LLMTime w/ Mistral | | Time-LLM w/ GPT-2 | | TimeGPT (2024) | | iTransformer (2024) | | TimesNet (2023) | |
|---|---|---|---|---|---|---|---|---|---|---|---|---|---|---|---|---|
| Metrcis | MSE | MAE | MSE | MAE | MSE | MAE | MSE | MAE | MSE | MAE | MSE | MAE | MSE | MAE | MSE | MAE |
| ETTh1 | 0.073 | 0.213 | 0.071 | 0.202 | 0.086 | 0.244 | 0.097 | 0.274 | 0.089 | 0.202 | 0.059 | 0.192 | 0.071 | 0.218 | 0.073 | 0.202 |
| w/ GWN | 0.077 | 0.219 | 0.076 | 0.213 | 0.087 | 0.237 | 0.094 | 0.291 | **0.102** | 0.231 | 0.059 | 0.193 | 0.072 | 0.216 | 0.074 | 0.202 |
| w/ DGA | **0.085** | **0.249** | **0.083** | **0.232** | **0.091** | **0.251** | **0.098** | **0.295** | 0.099 | 0.248 | **0.060** | **0.198** | **0.075** | **0.226** | **0.081** | **0.213** |
| ETTh2 | 0.263 | 0.372 | 0.155 | 0.267 | 0.237 | 0.373 | 0.277 | 0.492 | 0.238 | 0.361 | 0.161 | 0.297 | 0.171 | 0.296 | 0.166 | 0.316 |
| w/ GWN | 0.263 | 0.342 | 0.175 | 0.303 | 0.231 | **0.429** | 0.346 | 0.505 | 0.235 | 0.355 | 0.160 | 0.301 | **0.181** | 0.302 | 0.166 | 0.316 |
| w/ DGA | **0.275** | **0.408** | **0.201** | **0.327** | **0.257** | 0.425 | **0.356** | **0.554** | **0.302** | **0.441** | **0.171** | **0.312** | 0.179 | **0.308** | **0.169** | **0.321** |
| IstanbulTraffic | 0.837 | 0.844 | 0.805 | 0.779 | 0.891 | 1.005 | 0.826 | 0.973 | 0.995 | 1.013 | 1.890 | 1.201 | 1.081 | 0.995 | 1.095 | 1.022 |
| w/ GWN | 0.882 | 0.908 | 0.883 | 0.864 | 0.917 | 1.063 | 1.054 | 1.031 | 1.123 | 1.221 | 1.848 | 1.204 | **1.103** | 1.015 | 1.103 | 1.035 |
| w/ DGA | **0.955** | **1.073** | **1.417** | **1.214** | **0.994** | **1.083** | **1.744** | **1.217** | **1.161** | **1.328** | **1.918** | **1.218** | 1.097 | **1.034** | **1.155** | **1.047** |
| Weather | 0.005 | 0.051 | 0.004 | 0.048 | 0.008 | 0.072 | 0.006 | 0.057 | 0.004 | 0.034 | 0.004 | 0.043 | 0.005 | 0.053 | 0.003 | 0.042 |
| w/ GWN | 0.005 | 0.053 | 0.005 | 0.051 | 0.008 | 0.074 | **0.007** | **0.066** | 0.004 | 0.033 | 0.004 | 0.043 | **0.006** | 0.063 | 0.003 | 0.042 |
| w/ DGA | **0.006** | **0.063** | **0.006** | **0.061** | **0.009** | **0.079** | 0.007 | 0.062 | **0.005** | **0.052** | **0.006** | **0.071** | 0.006 | **0.065** | **0.004** | **0.045** |
| Exchange | 0.038 | 0.146 | 0.040 | 0.152 | 0.043 | 0.167 | 0.151 | 0.274 | 0.056 | 0.188 | 0.256 | 0.368 | 0.034 | 0.151 | 0.056 | 0.184 |
| w/ GWN | 0.042 | 0.179 | 0.046 | 0.182 | 0.050 | 0.185 | 0.160 | 0.298 | 0.059 | 0.194 | 0.329 | 0.413 | 0.044 | 0.166 | **0.065** | **0.195** |
| w/ DGA | **0.058** | **0.224** | **0.068** | **0.199** | **0.069** | **0.213** | **0.219** | **0.303** | **0.077** | **0.256** | **0.619** | **0.625** | **0.049** | **0.178** | 0.062 | 0.194 |

While we selected only three baseline LLM-based models for this study, the setup encompasses the primary approaches to LLM-based time series forecasting: pre-training a large model specifically for time series data (e.g., TimeGPT), leveraging well-developed general-purpose language models (e.g., LLMTime), and fine-tuning language models from other domains for time series forecasting (e.g., TimeLLM). This comprehensive selection provides a representative overview of the key strategies in adapting LLMs for time series tasks.

## 5.3 Experimental Procedures

We designed a series of experiments to evaluate the vulnerability of the baseline LLM models to adversarial attacks. For each model and dataset combination, we conducted the following procedures: (i) we applied targeted perturbations to the input data, carefully maintaining the overall structure of the original time series while subtly altering the data to mislead the LLMs' forecasting predictions; (ii) we introduced GWN with the same perturbation intensity; (iii) forecasting accuracy was measured using Mean Absolute Error (MAE) and Mean Squared Error (MSE), which allowed us to quantify the performance degradation caused by adversarial attacks compared to Gaussian noise.

## 5.4 Overall Comparison

As shown in Table 1, the experimental results demonstrate that the designed adversarial attacks significantly degraded forecasting performance across all datasets, as indicated by increased MSE and MAE values. Compared to GWN of the same perturbation intensity, our attacks had a much more detrimental effect on the models' predictions.

For TimeGPT, which is pre-trained with large-scale time series data, the adversarial attack led to a sharp rise in forecasting errors, demonstrating that even models specifically built for time series forecasting are vulnerable. For LLM-Time, which includes GPT-3.5, GPT-4, LLaMa, and Mistral as base models, the adversarial attack was even more pronounced. As illustrated in Figure 2, the attack caused a clear divergence between the forecasted values and the true time series, with all different variants of LLM-Time exhibiting larger deviations compared to GWN. GPT-3.5 and GPT-4, in particular, showed significant susceptibility, with their errors increasing substantially under adversarial conditions.

Across all models and datasets, the adversarial perturbations induced much greater disruptions than GWN, clearly impacting the predictions and demonstrating the precision of the attack in destabilizing LLM-based forecasting. This underlines the importance of developing robust defensive strategies to protect LLMs against such targeted adversarial attacks, as their current vulnerability poses a significant challenge for practical applications.

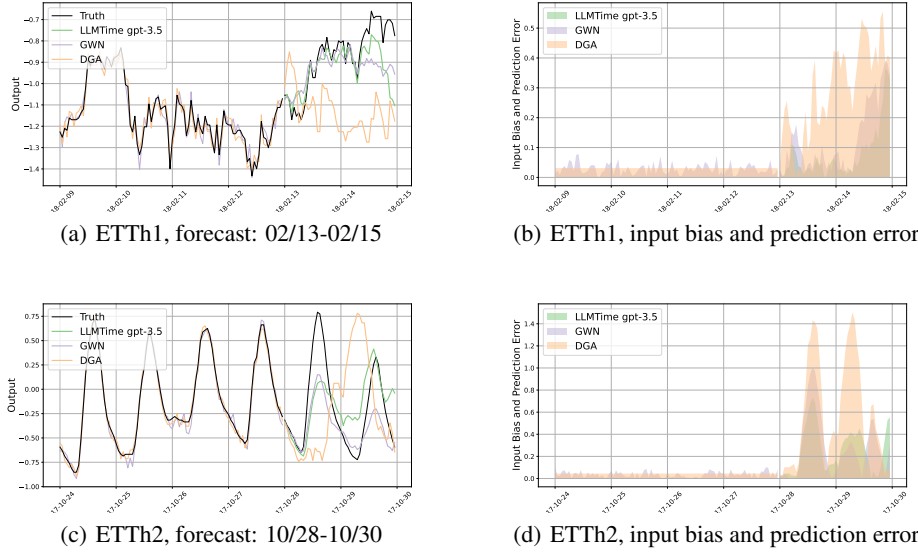

(a) ETTh1, forecast: 02/13-02/15

(b) ETTh1, input bias and prediction error

(c) ETTh2, forecast: 10/28-10/30

(d) ETTh2, input bias and prediction error

Figure 2: Prediction errors and input bias comparison for LLM-Time (with GPT-3.5) under adversarial attacks (DGA) and GWN. The figure highlights the greater disruption caused by DGA compared to GWN, showing significant deviations from the ground truth.

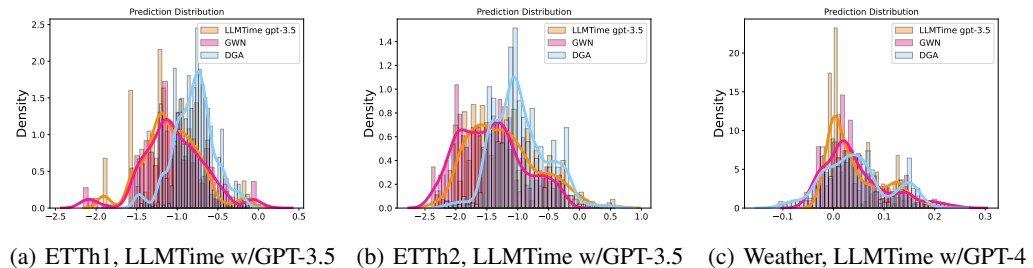

(a) ETTh1, LLMTime w/GPT-3.5  (b) ETTh2, LLMTime w/GPT-3.5  (c) Weather, LLMTime w/GPT-4

Figure 3: Prediction distribution comparison for LLM-Time (using GPT-3.5, GPT-4) across different datasets under clean input, GWN, and DGA.

## 5.5 Interpretation Study

Figure 3 illustrates the distribution shift in predictions caused by targeted perturbations on the LLM-based forecasting model. The proposed DGA method is designed to mislead the forecasting model, causing its predictions to resemble a random walk. As depicted in Figure 3, the "blue" shaded area, representing the perturbed prediction distribution, deviates significantly from the original "yellow" distribution and approaches a normal distribution. This shift underscores how subtle, well-crafted perturbations can manipulate the model into producing inaccurate forecasts. The effect of DGA-induced perturbations is pronounced when examining the prediction distributions, where errors are much more severe compared to the minor disruptions caused by GWN. These findings suggest that LLM-based forecasting models are highly susceptible to adversarial attacks that exploit the model's inherent vulnerabilities.

Additionally, the autocorrelation function (ACF) analysis provides further evidence of the detrimental impact of these adversarial attacks. Normally, LLMs demonstrate a strong ability to capture the temporal dependencies within time series data, maintaining coherent relationships between consecutive data points. However, as illustrated in Figure 4, when subjected to adversarial perturbations, these temporal dependencies break down, resulting in forecasts that no longer reflect the true underlying trends of the data. The disrupted autocorrelation patterns clearly illustrate the model's difficulty in

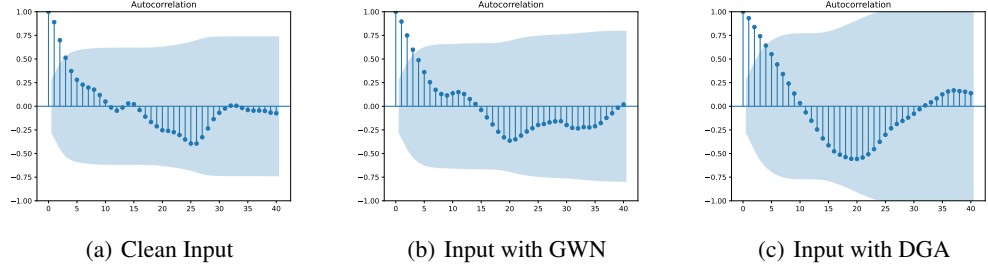

(a) Clean Input      (b) Input with GWN      (c) Input with DGA

Figure 4: Autocorrelation function curve comparison under clean input, GWN, and DGA on ETTh2 LLMTime w/GPT-3.5

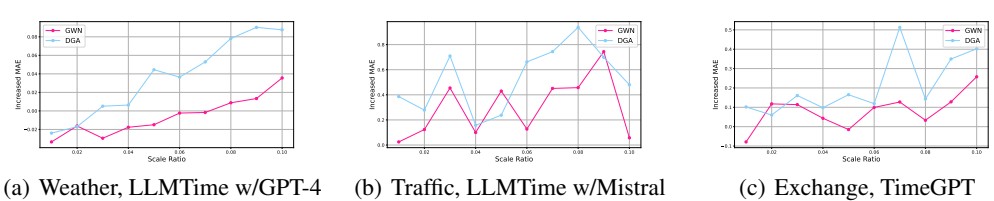

(a) Weather, LLMTime w/GPT-4    (b) Traffic, LLMTime w/Mistral    (c) Exchange, TimeGPT

Figure 5: Hyperparameter study on the effects of different scale ratios under GWN and DGA.

preserving the natural flow of time series data under attack. In contrast, the addition of Gaussian noise, though introducing some fluctuations, does not cause the same level of disruption, maintaining a closer relationship to the clean data.

### 5.6 Hyperparameter Study

We analyze the impact of varying scale ratios on model performance under both GWN and DGA adversarial attacks, with the vertical axis in Figure 5 representing the increase in MAE. This experiment was conducted across three different datasets using three LLM-based forecasting models. As demonstrated in the figure, DGA consistently results in a more significant increase in MAE compared to GWN as the scale ratio rises, indicating that DGA is more effective in disrupting the model's predictions. To balance imperceptibility and manipulation effectiveness, the perturbation scale can be chosen as 2% of the mean value of the given data.

## 6 Conclusion

In this study, we demonstrated the significant vulnerabilities of LLM-based models for time series forecasting to adversarial attacks. Through a comprehensive evaluation of TimeGPT and LLM-Time (with GPT-3.5, GPT-4, LLaMa, and Mistral as base models), we found that targeted adversarial perturbations, generated using Directional Gradient Approximation (DGA), caused substantial increases in prediction errors. These attacks were far more damaging than Gaussian White Noise (GWN) of similar intensity, highlighting the precision and effectiveness of the adversarial strategy.

The experimental results revealed that both large, pre-trained models like TimeGPT and fine-tuned models such as LLM-Time are highly susceptible to adversarial manipulation. The proposed attack can significantly degrade model performance across various datasets. This poses serious challenges for the deployment of LLMs in real-world time series applications, where reliability is critical.

Our findings emphasize the need for future research to focus on developing robust defense mechanisms to mitigate adversarial threats and enhance the resilience of LLM-based time series forecasting models. Without such protections, these models remain vulnerable to attacks that could undermine their practical utility in high-stakes environments. In addition, future studies should compare vulnerabilities of LLM with lighter models.

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
