# OpenReview forum: "Adversarial Vulnerabilities in Large Language Models for Time Series Forecasting"
_NeurIPS.cc/2024/Workshop/SafeGenAi — SafeGenAi Poster_

### Official Review · Reviewer_3zuj · 2024-10-08

**Rating:** 5
**Confidence:** 4

**Review:**

Summary: The paper proposes a gradient-free black-box attack that affects the output of LLM-based forecasting models and evaluates this vulnerability across several models and datasets. The findings indicate that LLMs are susceptible to adversarial manipulations and emphasizes the need for robust LLMs that can resist such attacks and enhance their reliability in real-world applications.

Strengths:
1. The writing is clear and readable, effectively introducing the analysis of adversarial attacks on LLM-based time series forecasting. It concisely communicates the vulnerabilities of such models and the approach for analyzing these attacks.
2. The paper proposes a gradient-free black-box attack that significantly degrades the prediction accuracy of LLM-based time series models, opening up new directions for future research on adversarial attacks in time series forecasting with LLMs.
3. The paper evaluates the proposed attack across multiple datasets and models, providing a solid empirical basis for the findings.

Weaknesses:
1. The paper focuses heavily on demonstrating the vulnerability of LLMs but does not offer actionable solutions or defense strategies, the contribution seems incomplete. It would be beneficial to include a discussion on potential ways to defend against these attacks or propose an experimental framework for mitigating their impact.

---

### Official Review · Reviewer_AsnJ · 2024-10-09
**Generally good**

**Rating:** 7
**Confidence:** 3

**Review:**

The paper explores the weaknesses of LLM-based time series forecasting by proposing a gradient-free black-box attack. The study employs both Gaussian White Noise (GWN) and Directional Gradient Approximation (DGA) to demonstrate that LLMs such as TimeGPT, GPT-3.5, GPT-4, and Mistral show severe vulnerabilities. The results confirm that LLMs are significantly impacted by adversarial attacks, calling for robust defense mechanisms.

## Quality

The paper provides a clear and coherent methodology for evaluating the adversarial vulnerabilities in time series forecasting using LLMs. The inclusion of five real-world datasets and two baseline models ensures the empirical relevance of the work. Additionally, the usage of two different perturbation strategies (GWN and DGA) strengthens the argument about the adversarial susceptibility of LLMs.

## Clarity

The paper is generally well-structured, and the proposed methodology is detailed. However, there are areas where more clarity would enhance comprehension. For example, the explanation of the DGA method, particularly the mathematical formulation, could be clearer for readers unfamiliar with gradient approximation techniques. Additionally, further details on the hyperparameter study could improve understanding of how DGA manipulates forecasting errors across different scale ratios.

## Originality

The originality of the paper lies in its novel focus on black-box attacks for time series forecasting, a largely under-explored area in the current LLM literature. While adversarial attacks on LLMs have been studied in text-based tasks, the application to time series forecasting is novel, offering new insights. However, the paper could have explored potential defenses, contributing further to the growing body of work on making LLMs more resilient in non-textual domains.

## Significance

This paper is significant for the field of time series forecasting with LLMs, especially given the increasing reliance on LLMs for real-world forecasting tasks. By showing that minimal perturbations can disrupt LLM predictions, it raises important concerns about the deployment of these models in high-stakes environments, such as finance or healthcare. The results suggest a critical gap in the security of these systems, making the findings relevant to both academic and industrial audiences.

## Pros:
- Introduces a novel adversarial attack method (DGA) for time series forecasting using LLMs.
- Comprehensive evaluation across multiple datasets and LLM architectures, demonstrating broad applicability.
- Strong experimental results showing clear degradation in forecasting performance under adversarial conditions.

Cons:
- Lack of detailed discussion on defensive strategies against such attacks, which would strengthen the paper’s practical relevance.
- The mathematical exposition of the proposed adversarial techniques, particularly DGA, could be simplified for broader readership.

The paper contributes valuable knowledge by exposing adversarial vulnerabilities in LLMs applied to time series forecasting. Although the absence of defenses weakens its practical utility, its methodological rigor and novel insights make it a good contender for acceptance.

---

### Official Review · Reviewer_4XgM · 2024-10-09
**The work investigates the vulnerabilities of Large Language Models (LLMs) in time series forecasting, particularly their susceptibility to adversarial attacks. The authors introduce a targeted adversarial attack framework that uses gradient-free and black-box optimization methods to generate minimal perturbations that significantly degrade forecasting accuracy across various datasets and LLM architectures. Despite LLMs' strong performance in time series tasks, their robustness and reliability are compromised by adversarial attacks, which introduce subtle changes leading to substantial prediction errors. Using five real-world datasets, the experiments show that even state-of-the-art models like TimeGPT are vulnerable to adversarial manipulations, emphasizing the need for robust defense mechanisms.**

**Rating:** 7
**Confidence:** 4

**Review:**

The paper presents a significant and timely investigation into the vulnerabilities of LLMs in the context of time series forecasting, which is an area that has seen limited exploration regarding adversarial attacks. The quality of the work is high, as it utilizes well-defined methodologies, including both gradient-free and black-box optimization methods to generate targeted adversarial perturbations. The experimental setup is robust, employing a diverse set of datasets and state-of-the-art LLM architectures, thus providing a comprehensive assessment of the proposed adversarial attack framework. In terms of clarity, the paper is generally well-structured and coherent, making it accessible to readers with varying levels of expertise. The introduction effectively sets the stage by contextualizing the importance of time series forecasting and the growing role of LLMs, while clearly outlining the gap in research regarding their adversarial vulnerabilities. The experimental procedures are described in detail, allowing for reproducibility, although some sections could benefit from clearer explanations of certain technical aspects related to the adversarial attack methodologies. The originality of the work is noteworthy, as it addresses a critical gap in the literature by applying adversarial attack frameworks specifically to LLMs used for time series forecasting. This is a fresh perspective that combines insights from NLP and time series analysis, enhancing the field's understanding of the inherent vulnerabilities of LLMs. The introduction of the Directional Gradient Approximation (DGA) as a method for generating adversarial perturbations is a particularly innovative contribution. The significance of the findings is profound. By demonstrating that LLMs, including those specifically designed for time series forecasting, are highly susceptible to adversarial attacks, the paper underscores the urgent need for robust defence mechanisms. This has important implications for the deployment of LLMs in real-world applications, particularly in high-stakes domains such as finance and healthcare, where reliable forecasting is crucial. Overall, this work not only advances academic understanding but also has practical implications for improving the resilience of LLM-based forecasting models.

The paper addresses a significant gap in the literature by investigating the adversarial vulnerabilities of LLMs in time series forecasting, an underexplored topic. The methodologies employed, including gradient-free and black-box optimization techniques, are innovative and provide a strong foundation for the proposed framework. The experimental design is robust, utilizing a diverse range of datasets and state-of-the-art LLM architectures, ensuring comprehensive evaluations of the proposed adversarial attacks. The results reveal critical insights into the susceptibility of LLMs to adversarial manipulations, emphasizing the need for developing defence mechanisms. The clarity of the writing and structure enhances the paper's accessibility, making it easier for readers from various backgrounds to engage with the content. The inclusion of real-world applications in the discussion underscores the practical significance of the findings, particularly for high-stakes environments.

Also, certain technical aspects of the adversarial attack methodologies could be better explained to improve clarity and understanding. Some readers may find the discussion of specific techniques, like Directional Gradient Approximation (DGA), to be lacking in detail, which might hinder reproducibility for those less familiar with the concepts. Additionally, the paper primarily focuses on the vulnerabilities of LLMs without exploring potential defence strategies in depth, which would enhance its impact. Finally, while the findings are significant, the study could benefit from a more extensive comparison with lighter models, as suggested in the conclusion, to provide a more comprehensive view of the landscape of model vulnerabilities.